# Application of Atmospheric-Pressure-Plasma-Jet Modified Flexible Graphite Sheets in Reduced-Graphene-Oxide/Polyaniline Supercapacitors

**DOI:** 10.3390/polym12061228

**Published:** 2020-05-28

**Authors:** Yu-Chuan Hao, Nurzal Nurzal, Hung-Hua Chien, Chen-Yu Liao, Fei-Hong Kuok, Cheng-Chen Yang, Jian-Zhang Chen, Ing-Song Yu

**Affiliations:** 1Department of Materials Science and Engineering, National Dong Hwa University, Hualien 97401, Taiwan; max850502@gmail.com (Y.-C.H.); nurzall@gmail.com (N.N.); 2Graduate Institute of Applied Mechanics, National Taiwan University, Taipei City 10617, Taiwan; R05543026@ntu.edu.tw (H.-H.C.); Ji3vu3nj@gmail.com (C.-Y.L.); R04543015@ntu.edu.tw (F.-H.K.); 3Department of Physics, National Dong Hwa University, Hualien 97401, Taiwan; athena1212624@gmail.com

**Keywords:** polyaniline, reduced graphene oxide, atmospheric pressure plasma, supercapacitor, graphite sheet

## Abstract

In this study, flexible and low-cost graphite sheets modified by atmospheric pressure plasma jet are applied to reduced-graphene-oxide/polyaniline supercapacitors. Surface treatment by atmospheric pressure plasma jet can make the hydrophobic surface of graphite into a hydrophilic surface and improve the adhesion of the screen-printed reduced-graphene-oxide/polyaniline on the graphite sheets. After the fabrication of reduced-graphene-oxide/polyaniline supercapacitors with polyvinyl alcohol/H_2_SO_4_ gel electrolyte, pseudo-capacitance and electrical double capacitance can be clearly identified by the measurement of cyclic voltammetry. The fabricated supercapacitor exhibits specific capacitance value of 227.32 F/g and areal capacitance value of 28.37 mF/cm^2^ with a potential scan rate of 2 mV/s. Meanwhile, the capacitance retention rate can reach 86.9% after 1000-cycle cyclic voltammetry test. A light-emitting diode can be lit by the fabricated reduced-graphene-oxide/polyaniline supercapacitors, which confirms that the supercapacitors function well and can potentially be used in a circuit.

## 1. Introduction

A conducting polymer is an organic polymer that can be used as a semiconductor or a conductor, which was discovered in 1977. Since then, conducting polymers have received much attention in the field of science and engineering [1]. Polyaniline (PANI) has advantages of high electrical conductivity, good specific capacitance, easy to synthesize, lower cost, and high chemical stability for versatile applications [2,3]. PANI has been widely applied for the energy storage devices, either as a conducting material or as an electroactive agent due to the tunable pseudocapacitive performance and its various oxidation states, especially employed in supercapacitors [4]. Their value of supercapacitance basically comes from the contribution of electrical double layer capacitance (EDLC) and/or Faradaic capacitance. Among them, EDLC is based on the adsorption and de-adsorption of electrolyte ions on the surfaces of electrodes, whereas Faradaic capacitance originates from the redox reaction of the surface materials in the electrodes [5]. Faradaic capacitance normally exists in metal oxides and conductive polymers [6,7]. PANI is one of the attractive candidate materials for Faradaic capacitance. However, PANI has some drawbacks, for instance low solubility or even insolubility in the most solvents, infusibility, and bad processability. The preparation of PANI nanocomposites combining with various materials has been developed to reduce the disadvantage. Graphene, a two-dimensional material, has the properties of ultra-high specific surface area, good electrical and thermal conductivity as well as excellent mechanical properties [8,9]; therefore, graphene and its derivatives have been applied in supercapacitors [10,11]. Specific capacitance in the value of 550 F/g could be realized theoretically if their surface area can be completely used. Until now, the liquid electrolyte supercapacitors with graphenes have reached the specific capacitance value of 100–300 F/g [12,13,14]. To introduce Faradaic capacitance into the device of supercapacitors, metal oxides and conductive polymers are typically used in the electrode materials. Therefore, by mixing PANI with graphenes, supercapacitors can take advantage of both EDLC and Faradaic capacitances [15,16]. Moreover, the supercapacitors with PANI nanocomposite inks can be fabricated by the screen-printing technique. Screen-printing is a very low-cost method in the industry, compatible with roll-to-roll manufacturing processes, and can be used on various types of substrates [17,18].

In the materials commonly used in the electrodes of supercapacitors, EDLC normally presents in carbonaceous materials. Among them, graphite sheets (GSs) have been widely used in a smartphone as a heat-spreading material in the behavior of high thermal conductivity [19]. Low-cost GSs have the properties of low density (2 g/cm^3^), high flexibility and excellent heat resistance. Besides, GSs have ultrahigh thermal conductivity (>1200 W/(m·K)), three times as high as copper (400 W/(m·K)) and six times as high as aluminum (250 W/(m·K)) [20,21]. Flexible graphite sheets with excellent thermal conductivity and chemical stability can be also used for an electrode of batteries, fuel cells and supercapacitors [22,23,24]. Meanwhile, the technique of atmospheric pressure plasma (APP) can be used for surface treatment of materials and operate at atmospheric pressure without any vacuum pumps and chambers. APP technique has been performed to be a highly economic and environmentally friendly tool for the material processing in a very large area. Plasmas sustained in the atmosphere are normally used to improve the adhesion and printing properties of polymers [25,26]. Nitrogen APP treatment can introduce the doping of nitrogen and improve the hydrophilicity of the carbonaceous nanocomposites to enhance the performance of gel-electrolyte supercapacitors [27].

Combining the advantages of these materials and techniques for the fabrication of supercapacitors, in this report, we propose the flexible graphite sheet modified by atmospheric pressure plasma jet (APPJ) before the screen-printing process with reduced-graphene-oxide/polyaniline (rGO/PANI) nanocomposite pastes. The characterizations of scanning electron microscopy (SEM), X-ray photoelectron spectroscopy (XPS) and water contact angle measurements perform the improvement of wettability between graphite sheets and rGO/PANI pastes. After the screen-printing process, rGO/PANI nanocomposites on APPJ-modified graphite sheets are also investigated. The manufacture of quasi-solid-state supercapacitors will be demonstrated. The performance of supercapacitors can also be enhanced via the evaluation of electrochemical analyses. Finally, a charged rGO/PANI supercapacitor will light a light-emitting diode up in the circuit.

## 2. Materials and Methods

The fabrication of APPJ-processed rGO/PANI supercapacitors is shown in Figure 1. First, the graphite sheet was cut and modified by scanning APPJ. The peak temperature of thermocouple under the scanning plasma jet was at the temperature of 550 °C. The operation parameters of APPJ were set at N_2_ flow rate of 42 standard liters per minute (slm), the operation voltage of 275 V, a repetition frequency of 25 kHz, the scan rate of 0.5 cm s^−1^ and the duration of 20 s to complete the modification [10]. The APPJ setup was described in detail elsewhere [28]. Then, the rGO/PANI pastes included 0.05 g PANI particles, 0.05 g rGO flakes, 1 g ethanolic solution containing 10 wt % ethyl cellulose, 0.811 g terpineol and 1.5 g ethanol, which were mixed and stirred for 24 h. Of the mixture 4 mL was condensed at 55 °C for 5 min. The rGO/PANI pastes were then screen-printed onto an APPJ-modified graphite sheet with an area of 2.0 cm × 1.5 cm. After the screen-printing process, it was annealed at 100 °C for the duration of 10 min as the electrode of supercapacitors [11]. Polyethylene terephthalate (PET) was used to fix the electrode. Then, a gel-electrolyte made of 1.5 g polyvinyl alcohol (PVA) and 15 mL 1 M H_2_SO_4_ was spread on the rGO/PANI electrode and then dried naturally for 24 h. This process was repeated for three times. Finally, the two pieces of rGO/PANI electrodes coated with PVA/H_2_SO_4_ were mechanically pressed face to face on the PVA/H_2_SO_4_ sides. This flexible quasi-solid-state supercapacitor with APPJ-treated graphite sheets was completed in the thickness of 2 mm.

For the characterization of materials, the rGO/PANI on APPJ-modified graphite sheets was conducted by SEM (Nova NanoSEM 230, FEI, Hillsboro, OR, USA) and XPS (VGS Thermo Scientific, Waltham, MA, USA) systems. The measurement of the water contact angle was conducted by using a goniometer (Model 100SB, Sindatek Instruments, Taipei, Taiwan). For the characterization of the supercapacitor, an electrochemical workstation (Zahner Zennium, Kronach, Germany) was employed for cyclic voltammetry (CV), galvanostatic charging/discharging (GCD) and cycling stability tests for 1000 cycles. CV of two electrodes system was performed from 0 to 0.8 V at 2, 20 and 200 mV·s^−1^, respectively. GCD was performed between 0 and 0.8 V at 0.5, 1 and 1.5 mA, respectively. The test of cycling stability was evaluated by CV measurements at the scanning rate of 200 mV·s^−1^ for 1000 cycles.

## 3. Results and Discussion

### 3.1. Surface Morphology of Flexible Graphite Sheets

Figure 2 shows the surface morphology of non-modified and APPJ-modified graphite sheets. In Figure 2a,c, some particles present on the flat surface before the surface modification by APPJ. After the APPJ modification, the particles on the graphite sheets became finer, and the surface of graphite sheets became rougher as shown in Figure 2b,d.

### 3.2. Surface Wettability of Graphite Sheets

Figure 3 performed the measurements of the water contact angle for non-modified and APPJ-modified graphite sheets. A high contact angle of 98.75° is shown on the non-treated graphite sheet. After the treatment of APPJ, the wettability could be significantly enhanced and the water droplet penetrated into the graphite sheet completely. APPJ surface treatment made the hydrophobic surface of graphite into a hydrophilic one. APPJ treatment could also improve the adhesion of the screen-printed rGO/PANI to graphite sheet, which could reduce the charge transfer impedance of supercapacitors and improve the performance of supercapacitors.

### 3.3. XPS Measurement of APPJ-Treated Graphite Sheets

In order to investigate the surface chemical composition of graphite sheets, the XPS survey scans of non-modified and APPJ-modified graphite sheets are shown in Figure 4a,b respectively. In the XPS spectrum of non-APPJ treated graphite sheet, the major peak of C-1s at the binding energy of 248.08 eV, and the minor peak of O-1s at the binding energy of 530.08 eV in Figure 4a. After the surface treatment by APPJ, the XPS spectrum of the APPJ-modified graphite sheet has a lower peak of C-1s, while the peak of O-1s is significantly enhanced and has an energy loss feature with a plasmon loss peak due to the interaction between the photoelectron and other electrons in Figure 4b. It is noted that a large amount of oxygen-containing functional groups are imported onto the surface of the graphite sheet. The plasma of APPJ is accompanied by excited-state particles such as UV or EUV short-wavelength light source, which can break the surface bonds. That is the reason that a large number of functional groups could be introduced into the surface of graphite sheets, and the oxygen-containing functional groups are effective in improving the surface hydrophilicity of the material. Therefore, it is assumed that the increase of surface roughness and the increase of oxygen-containing functional groups are the factors that greatly enhance the hydrophilicity of graphite sheets.

### 3.4. Surface Morphology for rGO/PANI Nanocomposites

Figure 5a,b show SEM images of as-deposited and 100 °C-annealed rGO/PANI nanocomposites on APPJ-modified graphite sheets, respectively. Before annealing, there were many lamellar structures and mixed with rGO. After annealing at 100 °C for 10 min, it was found that the lamellar structure disappeared, and the porous structure was clearly observed. It indicates that part of the organic binder is oxidized, gasified and diffused into the air, thus producing the rGO/PANI nanocomposites of the cloud-type porous structure.

Figure 6 shows SEM images of rGO/PANI nanocomposites after annealing in different magnifications. Rough and porous surface morphology can be observed clearly in Figure 6b,c. This porosity of rGO/PANI nanocomposites is useful for the penetration of the electrolyte in order to improve the contact area between the electrolyte and the electrode, which could perform a good capacitance value. Figure 6d shows clearly cloud-type polyaniline (PANI) was deposited on porous rGO.

### 3.5. XPS Analysis of rGO/PANI Nanocomposites on Graphite Sheets

Figure 7a,b show the XPS spectra of C-1s for as-deposited and 100 °C-annealed rGO/PANI, respectively. According to the semi-quantitative analysis by the peak fitting of XPS spectra, C-1s peak can be deconvoluted into C-C, C-N, C-O, C=O and O–C=O bonds at 284.5, 285.9, 286.5, 287.8 and 289.1 eV. These bonding configuration ratios of C-1s spectra can be also summarized in the figure [29]. The surface chemical composition of rGO/PANI nanocomposites on graphite sheets did not obviously change after annealing at 100 °C.

Moreover, the XPS spectra of O-1s for as-deposited and 100 °C-annealed rGO/PANI were performed in Figure 8a,b. The peak of O-1s can be deconvoluted into C=O, C-OH and COOH at the binding energy of 531.1, 532.3 and 533.3 eV [30]. The content of C=O and C-OH decreased a little as the annealing at 100 °C for 10 min. It indicates that part of terpineol and ethyl cellulose on the surface could be removed via evaporation after low-temperature annealing. Interestingly, we could find the COOH bond ratio increased by 7.89%. COOH is a hydrophilic functional group, which can increase the wettability between the gel electrolyte and the electrode material to get better performance of supercapacitors.

### 3.6. Electrochemical Test of rGO/PANI Nanocomposites on Graphite Sheets

In this section, we compared the electrochemical properties of non-printed rGO/PANI (graphite sheets as the electrode material) and 100 °C-annealed rGO/PANI supercapacitors. Figure 9 presents CV curves for (a) non-printed rGO/PANI and (b) 100 °C-annealed rGO/PANI supercapacitors under scan rates 200, 20 and 2 mVs^−1^, respectively. According to the measurements, Table 1 summarizes the areal and specific capacitances for these two supercapacitors. The values of areal capacitance (*C_A_*) and specific capacitance (*C_s_*) can be obtained by Equations (1) and (2) [31].
(1)$$C_{A}=\frac{1}{Av(V_{c}-V_{a})}\overset{V_{c}}{\underset{V_{a}}{\int }}I\left(V\right)dV$$
(2)$$C_{S}=\frac{1}{\;mv(V_{c\;}-V_{a})}\overset{V_{c}}{\underset{V_{a}}{\int }}I\left(V\right)dV$$
where “*I”* means the response current as a function of scanning potential *V*, *“ν”* means the potential scan rate, “*V_c_ − V_a_”* is the window of potential, “*m”* is the mass of the active substance, and “*A”* is the apparent area in the electrode. In Figure 9a, the rectangular part is mainly the electric double layer capacitors contributed by the surface of the graphite sheet. In Table 1, the specific capacitance of 0.35 F/g and areal capacitance of 0.044 mF/cm^2^ under a potential scan rate of 2 mV/s are both very low. However, Figure 9b shows the CV results of 100 °C-annealed rGO/PANI composed of a rectangle and a redox peak. The rectangle part of CV curves is mainly the electric double layer capacitor contributed by rGO, and the redox peak comes from the redox reaction of PANI. The specific capacitance and areal capacitance can reach 227.32 F/g and 28.37 mF/cm^2^ respectively under a potential scan rate of 2 mV/s. The specific capacitance decreased as the scanning rate increased as shown in Table 1. This is because the redox reaction requires a relatively long reaction time. The redox reaction in lack of response time at high scanning rates resulted in the decrease of capacitance. On the other hand, the electric-double layer capacitor at a higher scanning rate, ion reaction, was also less to follow the changes in the electric field. It will result in a decrease in capacitance value. The two factors cause specific capacitance to decrease as the scanning rate increased.

The GCD curves of 100 °C-annealed rGO/PANI supercapacitor are performed in Figure 10. The straight GCD curves can normally present the rapid charging and discharging characteristics for the fabricated supercapacitors. However, the GCD curves change their slopes at around 0.4 V, which shows the pseudocapacitive behavior [32]. The generation of external equivalent resistance resulted in a voltage drop, which was consistent to the results of the CV measurements in Figure 9b, where the redox peaks occurred. From the GCD curves, the areal capacitance (*C_A_*) and specific capacitance (*C_s_*) can be also calculated based on Equations (3) and (4) [33].
(3)$$C_{A}=\frac{I\cdot T}{A\cdot \Delta V}$$
(4)$$C_{S}=\frac{I\cdot T}{m\cdot \Delta V}$$
where “A” means the apparent area for the electrode, “m” is the mass of the active substance, “ΔV” is the potential window and *“I”* means the constant current.

According to the GCD curves, the areal and specific capacitances are summarized in Table 2. When the constant current decreased from 1.5 to 0.5 mA, the specific capacitance of 110.19 F/g and areal capacitance of 13.75 mF/cm^2^ increased to 126.72 F/g and 15.81 mF/cm^2^, respectively. This tendency of two capacitances agreed well with the results of CV measurements.

For the cycling stability test, Figure 11a shows the 1000-cycle CV results of the rGO/PANI supercapacitor with APPJ-modified graphite sheets, which was evaluated under a potential scanning rate of 200 mV/s. We can find the capacitance retention rate of 86.9% after 1000 cycles. For the charge of the supercapacitor, a 30 V battery was connected to two supercapacitors in series for 90 s. Additionally, then, the battery was quickly disconnected, and the charged supercapacitors were attached to a LED. Figure 11b shows the LED could be lit by a charged rGO/PANI supercapacitor. The charged supercapacitors could maintain the LED illuminate for around 1 min. This confirms that the manufactured rGO/PANI supercapacitors functioned well and could potentially be used in a circuit.

To compare the electrochemical performance with our previous work [10], the PANI/rGO supercapacitor using APPJ surface treatment of carbon cloth obtained the higher specific capacitance: *C_A_* 106.9 mF/cm^2^ and *C_s_* 580.2 F/g under the same potential scan rate of 2 mV/s. However, the supercapacitor with APPJ-modified graphite sheets has higher 1000-cycle cycling stability and could be in the lower fabrication cost.

## 4. Conclusions

We successfully fabricate rGO/PANI supercapacitors with APPJ-modified graphite sheets. The paste of rGO/PANI nanocomposites was screen-printed on the graphite sheets as the electrode of supercapacitor and annealed at the temperature of 100 °C. Form the electrochemical performance test, rGO/PANI nanocomposites can enhance the capacitance value of supercapacitors. In the cycle voltammetry test, the rectangle part comes mainly from the electric double layer capacitor by rGO, and the redox peak is contributed by the redox reaction of conducting polymer PANI. This supercapacitor exhibited specific capacitance of 227.32 F/g and areal capacitance of 28.37 mF/cm^2^ under a potential scanning rate of 2 mV/s. Interestingly, the capacitance retention rate could reach 86.9% after 1000-cycle CV stability test. Our experimental results also indicate that APPJ-modified graphite sheet is a promising substrate for supercapacitor applications, and the APPJ technique is a rapid and efficient tool that can improve the wettability, facilitating the penetration of electrolyte into rGO/PANI nanocomposites of supercapacitors.

## Figures and Tables

**Figure 1 polymers-12-01228-f001:**
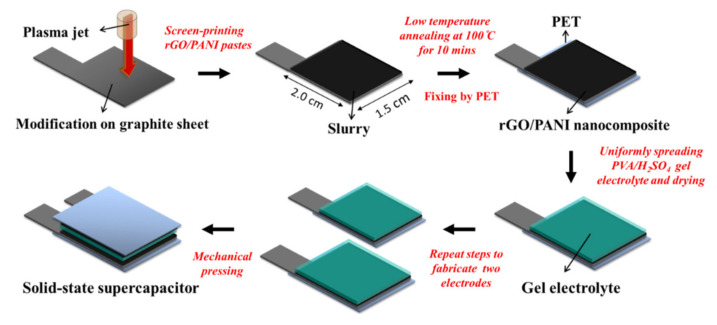
Process flow of atmospheric pressure plasma jet (APPJ)-treated reduced-graphene-oxide/polyaniline (rGO/PANI) supercapacitors.

**Figure 2 polymers-12-01228-f002:**
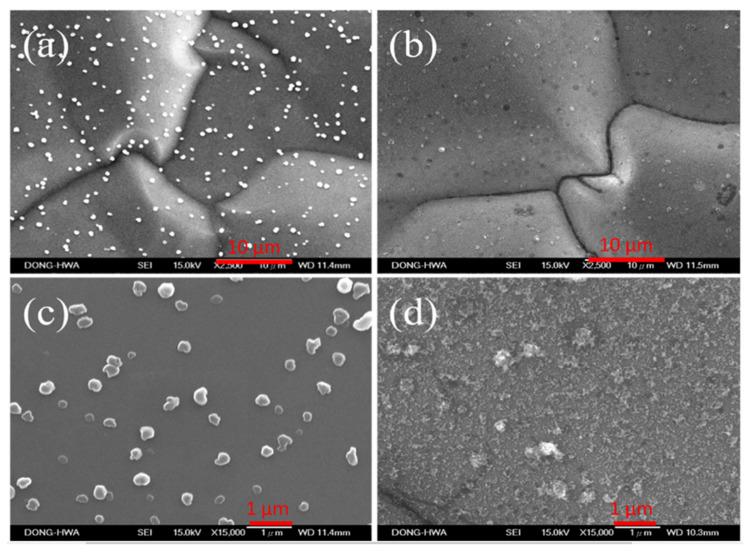
SEM images (2500×) of (**a**) non-modified and (**b**) APPJ-modified graphite sheets; SEM images in high magnification (15,000×) of (**c**) non-modified and (**d**) APPJ-modified graphite sheets.

**Figure 3 polymers-12-01228-f003:**
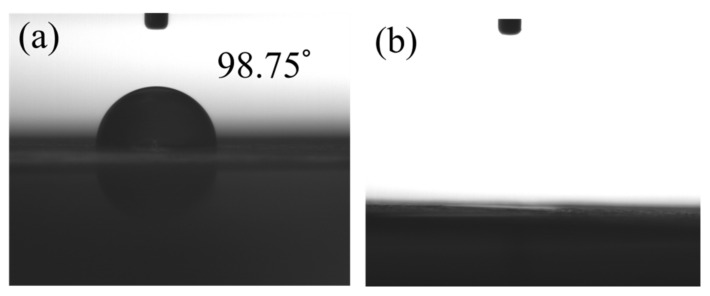
The images of contact angles measurements for (**a**) non-modified and (**b**) APPJ-modified graphite sheets.

**Figure 4 polymers-12-01228-f004:**
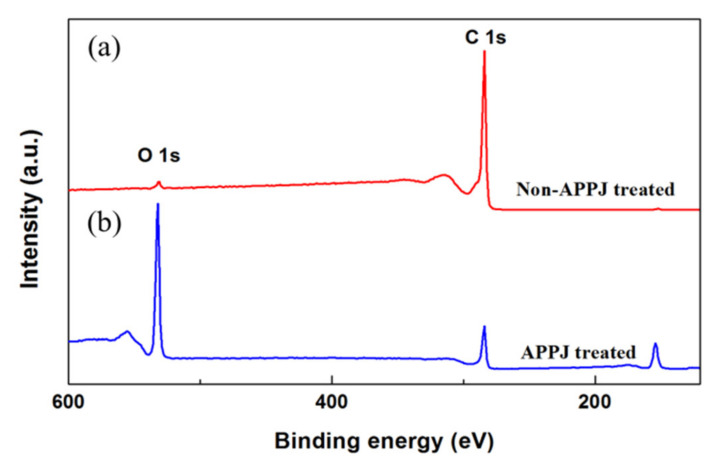
XPS survey spectra of (**a**) non-modified and (**b**) APPJ-modified graphite sheets.

**Figure 5 polymers-12-01228-f005:**
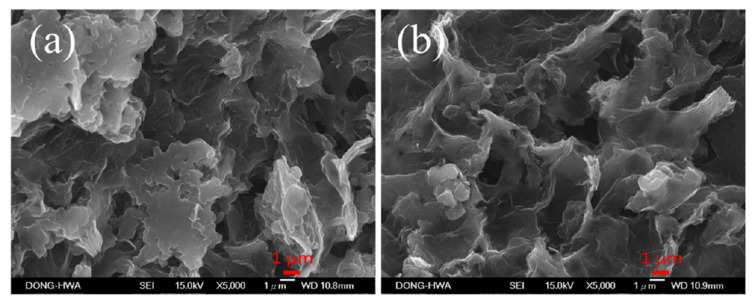
SEM images (5000×) of (**a**) as-deposited and (**b**) 100 °C-annealed rGO/PANI nanocomposites on APPJ-modified graphite sheets.

**Figure 6 polymers-12-01228-f006:**
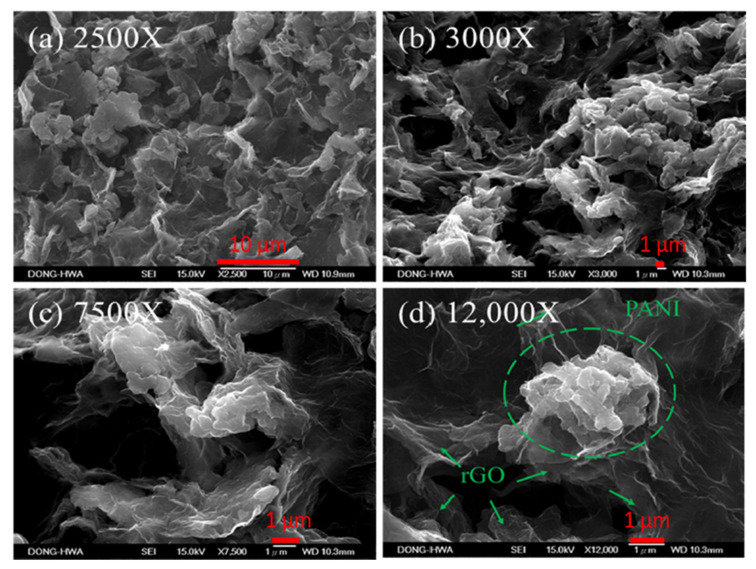
SEM images (**a**) 2500×, (**b**) 3000×, (**c**) 7500× and (**d**) 12,000× of 100 °C-annealed rGO/PANI nanocomposites on graphite sheets.

**Figure 7 polymers-12-01228-f007:**
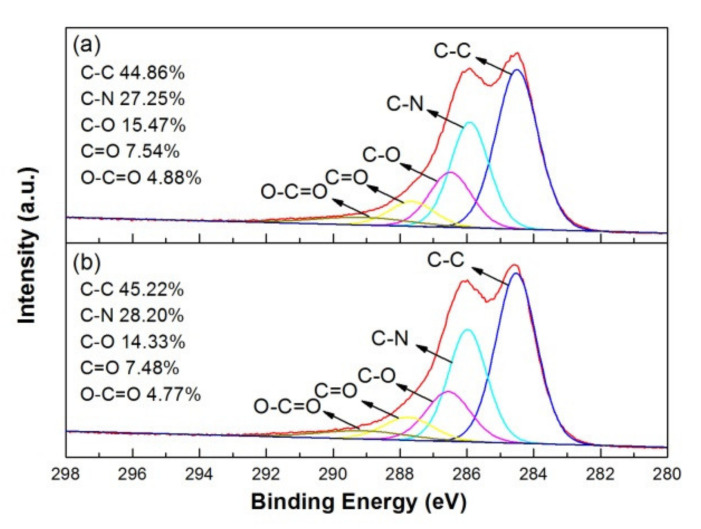
XPS C-1s spectrum and bonding ratio of (**a**) as-deposited and (**b**) 100 °C-annealed rGO/PANI nanocomposites.

**Figure 8 polymers-12-01228-f008:**
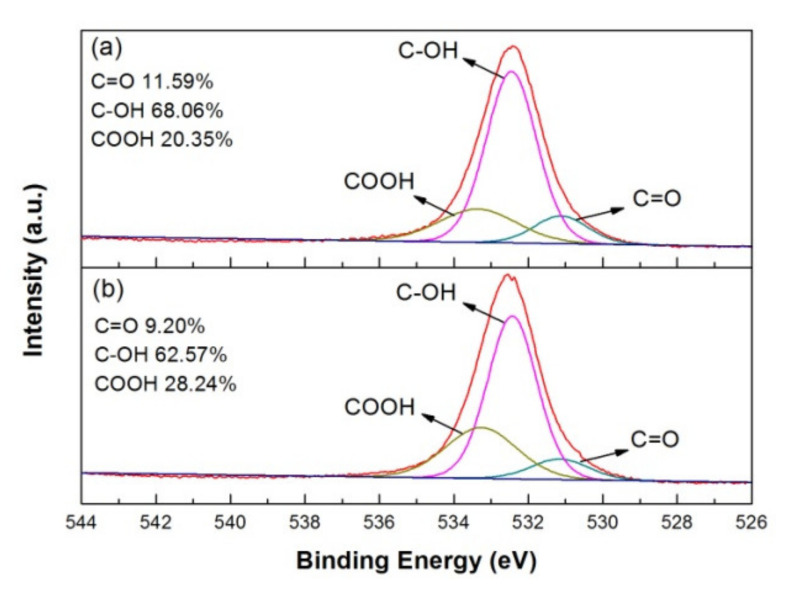
XPS O-1s spectrum and bonding ratio of (**a**) as-deposited and (**b**) 100 °C-annealed rGO/PANI nanocomposites.

**Figure 9 polymers-12-01228-f009:**
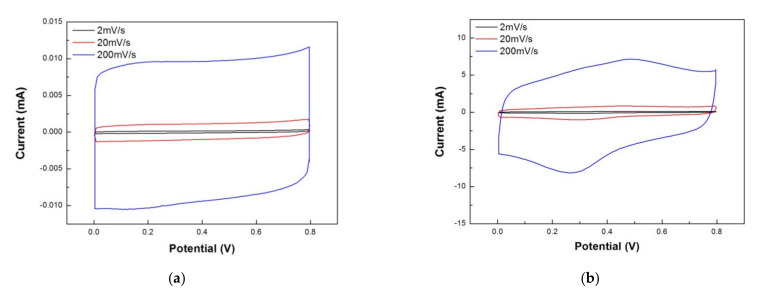
Cycle voltammetry results of (**a**) non-printed and (**b**) 100 °C-annealed rGO/PANI supercapacitors in the scanning rates of 200, 20 and 2 mVs^−1^, respectively.

**Figure 10 polymers-12-01228-f010:**
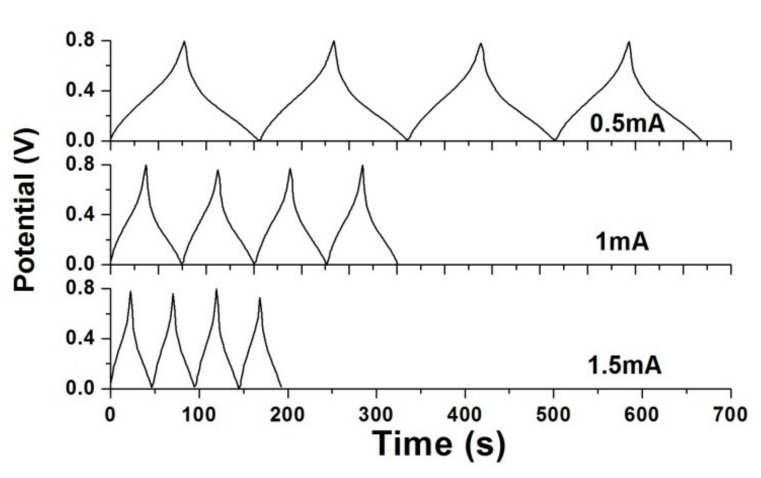
Galvanostatic charging/discharging (GCD) curves of 100 °C-annealed rGO/PANI supercapacitor.

**Figure 11 polymers-12-01228-f011:**
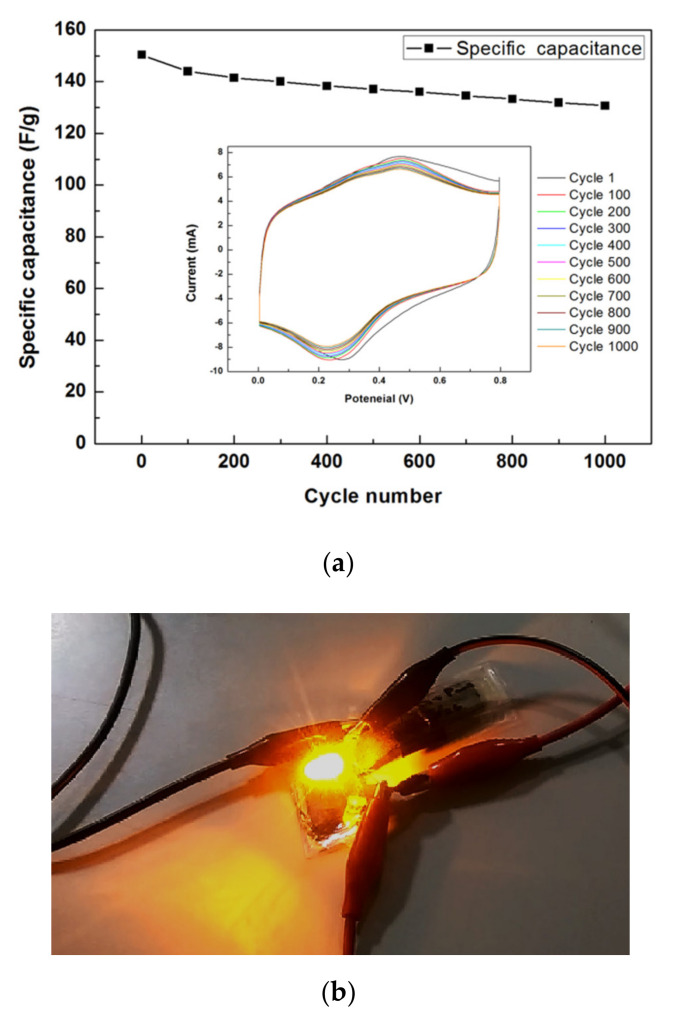
(**a**) 1000-cycle CV cycling stability test of rGO/PANI supercapacitor and (**b**) A LED powered by a charged rGO/PANI supercapacitor with APPJ-modified graphite sheets.

**Table 1 polymers-12-01228-t001:** Calculated areal and specific capacitance of non-printed and 100 °C-annealed rGO/PANI supercapacitors based on cyclic voltammetry (CV) curves.

	Non-Printed rGO/PANI		100 °C-Annealed rGO/PANI	
**Potential Scan Rate (mV/s)**	***C_A_*** **mF/cm^2^**	***C_s_*** **F/g**	***C_A_*** **mF/cm^2^**	***C_s_*** **F/g**
**2**	0.044	0.35	28.37	227.32
**20**	0.035	0.28	21.34	170.99
**200**	0.031	0.25	17.43	139.65

**Table 2 polymers-12-01228-t002:** Calculated areal and specific capacitance based on GCD curves.

	100 °C-Annealed rGO/PANI Supercapacitor	
**Constant Current (mA)**	***C_A_*** **mF/cm^2^**	***C_S_*** **F/g**
0.5	15.81	126.72
1	14.92	119.54
1.5	13.75	110.19

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
