# Peer review of "Application of Atmospheric-Pressure-Plasma-Jet Modified Flexible Graphite Sheets in Reduced-Graphene-Oxide/Polyaniline Supercapacitors"

_polymers, 2020, doi:10.3390/polym12061228_

Round 1
Reviewer 1 Report
The author only investigated one set of experiment conditions (such as the dosage of rGO and PANI solution, the condensation temperature, etc) in this paper. How do they determine this is the optimal fabrication parameters that gives the best cell performance? It would be more convincing if the impedance data will include and explain in relation to capacitance performance In the XPS spectra of Figure 4, the author should explain the additional peak near O1s in APPJ treated graphite sheets The conclusion section is not clear. The authors have to specify the salient features and technical advantages of the present system. In figure 11, please add other reported data of specific capacitance (rGO/PANI)). Then, the reader can compare your data with the previous data?
Author Response
Dear Reviewer,
Thanks for your thoughtful consideration of our manuscript. We also appreciate for the reviews and kind suggestions in order to make the manuscript more complete. We have tried to respond to the reviewer’s comments with a point-by-point response below and given a revised manuscript using the “Track Changes” function in Microsoft Word. We hope that our correction could meet with you approval.
Best regards,
Ing-Song YU
- Response to the reviewer 1:
-------------------------------------------------------------------------------------------------------
- The author only investigated one set of experiment conditions (such as the dosage of rGO and PANI solution, the condensation temperature, etc) in this paper. How do they determine this is the optimal fabrication parameters that give the best cell performance?
Response: Thanks for the comment from reviewer 1. In the report, we focus more on the fabrication of the supercapacitors using different materials (rGO/PANI and graphite sheets) and techniques (APPJ and screen-printing). Here, we have the characterizations of those materials and demonstrate the initial electrochemical performance of the supercapacitors. For the optimization of annealing temperature for rGO/PANI, that has been reported in the reference 11. For other fabrication parameters, the optimization of the supercapacitors will need more experiments to have statistic and reliability analyses, which will be interesting and important for the future applications of this supercapacitor.
[11] Liao, C.Y.; Chien, H.H.; Hao, Y.C.; Chen, C.W.; Yu, I.S.; Chen, J.Z. Low-temperature-annealed reduced graphene oxide-polyaniline nanocomposites for supercapacitor applications. J. Electron. Mater. 2018, 47, 3861-3868.
It would be more convincing if the impedance data will include and explain in relation to capacitance performance.
Response: Thanks for the great suggestion. Due to the limitation of samples, we did not have the measurements of electrochemical impedance spectroscopy (EIS). However, in our previous work (reference 10), the EIS does indicate a decreasing charge-transfer at the electrode/electrolyte interface for supercapacitors with APPJ treatment of carbon cloth.
[10] Chien, H.H.; Lia, C.Y.; Hao, Y.C.; Hsu, C.C.; Cheng, I.C.; Yu, I.S.; Chen, J.Z. Improved performance of polyaniline/reduced-graphene-oxide supercapacitor using atomospheric-pressure-plasma-jet surface treatment of carbon cloth. Electrochimi. Acta 2018, 260, 391-399.
In the XPS spectra of Figure 4, the author should explain the additional peak near O1s in APPJ treated graphite sheets.
Response: Thanks for the comment. During the survey of XPS spectra, a big energy loss feature with plasmon loss peak is often observed to higher binding energy of the main O-1s peak due to the interaction between the photoelectron and other electrons. Please see the revised manuscript in lines 392-394.
The conclusion section is not clear. The authors have to specify the salient features and technical advantages of the present system.
Response: Thanks for the reminding to let the manuscript better. We have modified the conclusion. Please see the revised manuscript in lines 525-535.
In figure 11, please add other reported data of specific capacitance (rGO/PANI)). Then, the reader can compare your data with the previous data?
Response: Thanks for the comments. We have compared the data of specific capacitance and cycling stability test with our previous work in the reference 10. Please see the revised manuscript in lines 493-497.
Reviewer 2 Report
Authors reported an interesting study about the fabrication of supercap using rGO and PANI. Also they reported the use of plasma treatment to enhance the performance of the materials. Tha paper is quite interesting but it presents some minor and major flasw summurazied as follow:
lines 16-26: Do not use acronyms in the abstract
line 119: The scale on figure 2 are unreadable. With this resolution you cannot infer nothing. PLease modify the scales because the magnification used are not sufficent.
lines 136-150: The XPS interpretation is far to be sufficent and exhaustive. You must interpolate the signals and after you can evalute.
line 151-171: Figure 5 and 6 hae the same proble of figure 2. This paragraph suffers of the same sma problem
line 169: How could you say that is rGO?
line 225: Data of table 1 are reported without errors. In this condition you cannot discrimiante from each other. Reporting data without erros is a serious methodological flaw.
Line 247: Same as table 1.
The absence of errors and staticall analysis of the data is great problem because without erros data are just merely numbers.
Considering the points raised above, i suggest major revisiosn for this paper.
Author Response
Dear Reviewer,
Thanks for your thoughtful consideration of our manuscript. We also appreciate for the reviews and kind suggestions in order to make the manuscript more complete. We have tried to respond to the reviewer’s comments with a point-by-point response below and given a revised manuscript using the “Track Changes” function in Microsoft Word. We hope that our correction could meet with you approval.
Best regards,
Ing-Song YU
- Response to the reviewer 2:
-------------------------------------------------------------------------------------------------------
General comment
Authors reported an interesting study about the fabrication of supercapacitors using rGO and PANI. Also they reported the use of plasma treatment to enhance the performance of the materials. The paper is quite interesting but it presents some minor and major flaw summarized as follow:
Our response
We are thankful to the reviewer 2 for the effort and valuable time, and also we thank the reviewer’s recognition for our work about the study of supercapacitors. Note that all changes mentioned below are marked in the revised manuscript, and the point-by-point responses are in the followings:
lines 16-26: Do not use acronyms in the abstract.
Response: Thanks for the comment. We have modified the abstract without any acronyms. Please see the revised manuscript in lines 17-28.
line 119: The scale on figure 2 are unreadable. With this resolution you cannot infer nothing. Please modify the scales because the magnifications used are not sufficient.
Response: Thanks for the comment. Figure 2 has been modified.
lines 136-150: The XPS interpretation is far to be sufficient and exhaustive. You must interpolate the signals and after you can evaluate.
Response: Thanks for the suggestion. More interpretation about XPS measurement has been added in the revised manuscript in lines 273- 300.
line 151-171: Figure 5 and 6 have the same problem of figure 2. This paragraph suffers of the same problem
Response: Thanks for the comment. Figure 5 and Figure 6 have been modified.
line 169: How could you say that is rGO?
Response: Thanks for the comment. Our raw materials came from the mixture of PANI particles and rGO flakes, so we can roughly indentify them from the SEM images. Meanwhile, the conductivity of PANI and rGO are different, so we can see more charging effect of PANI particles under the observation of SEM.
line 225: Data of table 1 are reported without errors. In this condition you cannot discrimiante from each other. Reporting data without erros is a serious methodological flaw.
Line 247: Same as table 1
The absence of errors and statically analysis of the data is great problem because without erros data are just merely numbers.
Response: Thanks for the comment. As the reviewer said the statistic analysis of data is important for the scientific report. However, in the report, we focus more on the fabrication of the supercapacitors using different materials and techniques, the characterizations of these materials, and demonstrating the initial electrochemical performance of the supercapacitors. Thanks for the good suggestion; the more fabricated supercapacitors to have statistic and reliability analyses will be interesting and important for the future applications of this supercapacitor.